# EGCG Prevents the Onset of an Inflammatory and Cancer-Associated Adipocyte-like Phenotype in Adipose-Derived Mesenchymal Stem/Stromal Cells in Response to the Triple-Negative Breast Cancer Secretome

**DOI:** 10.3390/nu14051099

**Published:** 2022-03-05

**Authors:** Narjara Gonzalez Suarez, Yuniel Fernandez-Marrero, Sima Torabidastgerdooei, Borhane Annabi

**Affiliations:** 1Laboratoire d’Oncologie Moléculaire, Département de Chimie, and CERMO-FC, Université du Québec à Montréal, Montreal, QC H3C 3P8, Canada; gonzalez_suarez.narjara@courrier.uqam.ca (N.G.S.); torabidastgerdooei.sima@courrier.uqam.ca (S.T.); 2Biological Sciences Platform, Sunnybrook Research Institute, Sunnybrook Health Science Centre, Toronto, ON M4N 3M5, Canada; yuniel.fernandez@sunnybrook.ca

**Keywords:** pre-adipocytes, EGCG, EMT, Snail, IL-6, TNBC

## Abstract

Background: Triple-negative breast cancer (TNBC) cells secretome induces a pro-inflammatory microenvironment within the adipose tissue, which hosts both mature adipocytes and adipose-derived mesenchymal stem/stromal cells (ADMSC). The subsequent acquisition of a cancer-associated adipocyte (CAA)-like phenotype is, however, unknown in ADMSC. While epidemiological studies suggest that consuming a polyphenol-rich diet reduces the incidence of some obesity-related cancers, the chemopreventive impact of green tea-derived epigallocatechin-3-gallate (EGCG) against the cues that trigger the CAA phenotype remain undocumented in ADMSC. Methods: Human ADMSC were exposed to human TNBC-derived MDA-MB-231 conditioned media (TNBC cells secretome) supplemented or not with EGCG. Differential gene expression was assessed through RNA-Seq analysis and confirmed by RT-qPCR. Protein expression levels and the activation status of signal transduction pathways mediators were determined by Western blotting. ADMSC chemotaxis was assessed by a real-time cell migration assay. Results: The TNBC cells secretome induced in ADMSC the expression of the CAA cytokines CCL2, CCL5, IL-1β, and IL-6, and of immunomodulators COX2, HIF-1α, VEGFα, and PD-L1. The epithelial-to-mesenchymal biomarker Snail was found to control the CAA phenotype. EGCG inhibited the induction of CAA genes and the activation status of Smad2 and NF-κB. The induced chemotactic response was also inhibited by EGCG. Conclusion: The induction of an inflammatory and CAA-like phenotype in ADMSC can be triggered by the TNBC cells secretome, while still efficiently prevented by diet-derived polyphenols.

## 1. Introduction

The acquisition of a cancer-associated adipocyte (CAA) phenotype can be viewed as an adaptive characteristic of cells residing within the adipose tissue and that respond to cues that originate from the tumor microenvironment (TME) [1,2]. Accordingly, adipocytes tumor infiltration has been reported in breast, ovarian, colorectal, and pancreatic cancers, ultimately leading to the instauration of a pro-inflammatory state that promotes carcinogenic events in return [3,4]. Such pro-malignancy role of the adipose tissue has been described primarily in breast cancer, where the residing adipocytes represent the most abundant stromal cell type and constitute the main source of pro-inflammatory cytokines and growth factors [5,6,7].

The CAA phenotype characterizes adipose tissue-derived cells with morphologically smaller and irregular sizes, as well as with decreased content and dispersed pattern of lipid droplets [8]. These cells also present an activated state attributable to the overexpression and secretion of the chemokine (C-C motif) ligand 2 (CCL2, also known as MCF-1), the chemokine (C-C motif) ligand 5 (CCL5, also known as RANTES), inflammatory cytokines including interleukin (IL)-1β, IL-6, tumor necrosis factor (TNF)-α, and matrix metalloproteinase (MMP)-11 [6]. The CAA phenotype further associates with increased releases of metabolites such as lactate, pyruvate, free fatty acids, and ketone bodies [9]. Such adaptive metabolic state is believed to mimic the hypoxic status and to contribute to immunosuppressive events within the TME, in part through the upregulation of hypoxia-inducible factor-1α (HIF-1α) and c-Myc [10,11,12]. In terms of TME localization, these cells have been ascribed to the invasive front of human breast cancer tumors [6,13,14].

Epidemiological studies have implied the existence of an association between excess adipose tissue and a higher incidence/progression of breast cancer [15,16]. In obesogenic conditions, the excessive expansion in adipose tissue triggers a chronic low-grade inflammation state recognized to create an environment that can sustain tumoral progression [17]. Therefore, a dynamic crosstalk between resident adipocytes and paracrine response to TME signals appears to play a crucial role in the onset of an aggressive tumor phenotype [18,19]. Epidemiological studies indicate that consumption of a polyphenol-rich diet reduces the incidence of obesity-related cancers [20,21]. Nevertheless, the cues triggering the CAA phenotype remain less understood at the early stages of adipocyte maturation as well as in adipocyte-derived mesenchymal stem/stromal cells (ADMSC, also referred to as pre-adipocytes [22]). Those cells have been demonstrated to have the ability to differentiate into mesodermal tissue lineages including adipose through the regulation of key transcriptional factors involved in early adipogenesis [22].

Among the phytochemicals targeting adipogenesis, the green-tea-derived epigallocatechin 3-gallate (EGCG) prevented the acquisition of a more invasive phenotype in a triple-negative breast cancer (TNBC)-derived MDA-MB-231 cell model, triggered by the secretome of mature adipocytes but not by the secretome from human ADMSC [23]. The adipose tissue microenvironment also promoted TNBC cell invasiveness and dissemination by producing CCL5 [24]. ADMSC were also suggested to promote progression and metastatic spread in breast cancer by switching to a more malignant phenotype, leading to a worse prognosis [25]. The sum of this evidence supports the concept that diet-derived phytochemicals could prevent the onset of an inflammatory obesogenic environment that favors the acquisition of a CAA-like phenotype.

To unveil the key mediators in the crosstalk between cancer cells and resident adipose tissue cells, studies have so far mostly characterized the promoting role of adipocytes in tumor progression [26]. However, fewer have focused on how the TME-mediated reshaping of preadipocytes or ADMSC, or how dedifferentiated mature adipocytes arise. Therefore, the present study aims at characterizing how soluble factors secreted from the TNBC-derived MDA-MB-231 cell line can mediate the acquisition of an inflammatory and CAA phenotype, and the chemotactic response in ADMSC. Finally, the chemopreventive impact of EGCG was assessed as a model for nutraceutical intervention against the acquisition of a CAA-like phenotype in ADMSC.

## 2. Materials and Methods

### 2.1. Materials

Sodium dodecylsulfate (SDS), epigallocatechin-3-gallate (EGCG) and bovine serum albumin (BSA) were purchased from Sigma-Aldrich Canada (Oakville, ON, Canada). Electrophoresis reagents were purchased from Bio-Rad (Mississauga, ON, Canada). The enhanced chemiluminescence (ECL) reagents were from Amersham Pharmacia Biotech (Baie d’Urfé, QC, Canada). Micro bicinchoninic acid protein assay reagents were purchased from Pierce (Rockford, IL, USA). The polyclonal antibodies against Snail, Slug, phosphorylated and total NF-κβ (p105), Smad2, and STAT3 were obtained from Cell Signaling Technology Inc. (Danvers, MA, USA). Monoclonal antibody (mAb) against human IL-6 was purchased from (New England Biolabs Ltd., Whitby, ON, Canada), rabbit IgG isotype control was obtained from Abcam (Cat ab172730, clone EPR25A); and a mAb against glyceraldehyde 3-phosphate dehydrogenase (GAPDH) was from Advanced Immunochemical Inc. (Long Beach, CA, USA). Horseradish peroxidase-conjugated donkey anti-rabbit and anti-mouse IgG secondary antibodies were obtained from Jackson ImmunoResearch Laboratories (West Grove, PA, USA). Protein A sepharose beads were obtained from GE Healthcare (Uppsala, Sweden). Gelatin was obtained from Sigma Aldrich (Oakville, ON, Canada).

### 2.2. Cell Culture and TNBC Cells Secretome Collection

The human adipose-derived mesenchymal stem/stromal cells (ADMSC) and TNBC-derived cell line MDA-MB-231 were purchased from the American Type Culture Collection (ATCC, Manassas, VA, USA). ADMSC were grown in Mesenchymal Stem Cell Basal Medium (ATCC, PCS-500-030) and supplemented with Mesenchymal Stem Cell Growth Kit—Low Serum (ATCC, PCS-500-040). They were further reported to have the capacity to undergo adipogenesis [27]. MDA-MB-231 were grown in EMEM Medium (Wisent, 320-036-CL) supplemented with 10% of fetal bovine serum. All cells were cultured at 37 °C under a humidified 95–5% (*v*/*v*) mixture of air and CO_2_. The TNBC cells secretome was generated upon a 48 h serum deprivation of a ~70% confluent MDA-MB-231 culture. Next, the conditioned media (CM) was harvested and centrifuged at 1500× *g* for 20 min to eliminate cell debris. CM was aliquoted and kept at −20 °C. To evaluate the induction of the CAA phenotype, ADMSC were cultured with the TNBC cells secretome in the presence or absence of 10 μM EGCG for 24 h. Then, cells were collected for total RNA extraction, protein isolation, or cell migration studies.

### 2.3. Total RNA Isolation, cDNA Synthesis, and Real-Time Quantitative PCR

Total RNA was extracted from cell monolayers using 1 mL of TriZol reagent for a maximum of 3 × 10^6^ cells as recommended by the manufacturer (Life Technologies, Gaithersburg, MD, USA). For cDNA synthesis, 1–2 µg of total RNA was reverse-transcribed using a high-capacity cDNA reverse transcription kit (Applied Biosystems, Foster City, CA, USA) or, in the case of the gene array, R2 First Strand kit (QIAGEN, Valencia, CA, USA). The cDNA was stored at −80 °C prior to PCR. Gene expression was quantified by real-time quantitative PCR using iQ SYBR Green Supermix (Bio-Rad, Hercules, CA, USA). DNA amplification was carried out using an Icycler iQ5 (Bio-Rad), and product detection was performed by measuring binding of the fluorescent dye SYBR Green I to double-stranded DNA. The following primer sets were from QIAGEN: Snail (Hs_SNAI1_1_SG, QT00010010), IL-6 (Hs_IL6_1_SG, QT00083720), RPS18 (Hs_RPS18_2_SG, QT02323251) and PPIA (Hs_PPIA_4_SG, QT01866137). The relative quantities of target gene mRNA were normalized against internal housekeeping genes PPIA and RPS18. The RNA was measured by following a ∆C_T_ method employing an amplification plot (fluorescence signal vs. cycle number). The difference (∆C_T_) between the mean values in the triplicate samples of target gene and the housekeeping genes was calculated with the CFX manager Software version 2.1 (Bio-Rad), and the relative quantified value (RQV) was expressed as 2^−∆CT^.

### 2.4. Total RNA Library Preparation and Sequencing

Total RNA (500 ng) was used for library preparation. RNA quality control was assessed with the Bioanalyzer RNA 6000 Nano assay on the 2100 Bioanalyzer system (Agilent technologies, Mississauga, ON, Canada), and all samples had an RNA integrity number (RIN) above eight. Library preparation was carried out with the KAPA mRNA-Seq HyperPrep kit (KAPA, Cat no. KK8581). Ligation was made with Illumina dual-index UMI, and 10 PCR cycles were required to amplify cDNA libraries. Libraries were quantified by QuBit and BioAnalyzer DNA1000. All libraries were diluted to 10 nM and normalized by qPCR using the KAPA library quantification kit (KAPA; Cat no. KK4973). Libraries were pooled to equimolar concentrations. Three biological replicates were generated. Sequencing was performed with the Illumina Nextseq500 using the Nextseq High Output 75 (1 × 75 bp) cycles kit. Around 15–20 M single-end PF reads were generated per sample. Library preparation and sequencing was performed at the Genomic Platform of the Institute for Research in Immunology and Cancer (IRIC, Montreal, QC, Canada).

### 2.5. Reads Alignment and Differential Expression Analysis

Reads were 30 trimmed for quality and adapter sequences using Trimmomatic version 0.35, and only reads with at least 50 bp in length were kept for further analyses. Trimmed reads were aligned to the reference human genome version GRCh38 (gene annotation from Gencode version 37, based on Ensembl 103) using STAR version 2.7.1a [28]. Gene expressions were obtained both as read count directly from STAR and computed using RNA-Seq by Expectation Maximization (RSEM) [29] to obtain normalized gene and transcript-level expression, in TPM values, for these stranded RNA libraries. Differential expression analysis was performed using DESeq2 version 1.22.2 [30]. The package limma [31] was used to normalize expression data and read counts data were analyzed using DESeq2. Principal component analysis (PCA) for the first two most significant components was conducted with R packages [32] found in iDEP (integrated Differential Expression and Pathway) analysis [33]. iDEP was also used to determine significant differentially expressed genes (DEGs) with DESeq2 false discovery rate (FDR) adjusted *p*-value of 0.05 and fold-change with a cutoff of two.

### 2.6. Gene Ontology Pathway Enrichment Analysis

The genes were filtered by absolute fold change (FC) and FDR (|FC| > 2, FDR < 0.001) and then used for pathway enrichment analysis on active subnetworks prepared using the library pathfindR [34]. Genes common to samples treated with EGCG or vehicle were z-normalized and clustered using a consensus from ten independent k-means runs. The results were visualized as a heatmap using the package ComplexHeatmap [35]. All analyses were performed using R version 4.1.1 (10 August 2021). Gene ontology (GO) enrichment analysis for protein class and molecular function of the genes in clusters 5–7 were performed using the GO online resource (geneontology.org). The enrichment of the upregulated genes per cluster was determined using all genes detected in the control sample as background. An FDR < 0.05 was used as cut off.

### 2.7. Human Cancer Inflammation and Immunity Crosstalk PCR Array

The RT^2^ Profiler^TM^ PCR Array for Human Cancer Inflammation and Immunity Crosstalk (PAHS-181Z) was used according to the manufacturer’s protocol (QIAGEN). The detailed list of the key genes assessed can be found on the manufacturer’s website (https://geneglobe.qiagen.com/us/product-groups/rt2-profiler-pcr-arrays; accessed on 13 January 2022). Using real-time quantitative PCR, we reliably analyzed the expression of a focused panel of genes related to the inflammatory response, including some of the cancer-associated adipocytes markers already published. Relative gene expression was calculated using the 2^−∆∆CT^ method (“delta-delta” method), in which C_T_ indicates the fractional cycle number where the fluorescent signal crosses the background threshold. This method normalizes the ∆C_T_ value of each sample, using five housekeeping genes (B2M, HPRT1, RPL13A, GAPDH, and ACTB). The normalized FC values are then presented as average FC = 2 (average ^∆∆^C^T^). To minimize false positive results, only genes amplified less than 35 cycles were analyzed. The resulting raw data were then analyzed using the PCR Array Data Analysis Template (http://www.sabiosciences.com/pcrarraydataanalysis.php; accessed on 13 January 2022). This integrated web-based software package automatically performs all ∆∆C_T_-based FC calculations from the uploaded raw thresholded cycle data.

### 2.8. RNA Interference

Cells were transiently transfected with siRNA using Lipofectamine-2000 transfection reagent (Thermo Fisher Scientific, Waltham, MA, USA). Gene silencing was performed using 20 nM siRNA against SNAIL (Hs_SNAI1_5 HP siRNA, SI02636424), IL-6 (Hs_IL6_1 siRNA, SI00012572) or scrambled sequences (AllStar Negative Control siRNA, 1027281). The above small interfering RNA and mismatch siRNA were all synthesized by QIAGEN and annealed to form duplexes.

### 2.9. Western Blot

Cells were lysed in a buffer containing 1 mM each of NaF and Na_3_VO_4_, and proteins (10–20 µg) were separated by SDS-polyacrylamide gel electrophoresis (PAGE). Next, proteins were electro-transferred to polyvinylidene difluoride membranes and blocked for 1 h at room temperature with 5% nonfat dry milk in Tris-buffered saline (150 mM NaCl, 20 mM Tris-HCl, pH 7.5) containing 0.3% Tween-20 (TBST; Bioshop, TWN510-500). Membranes were washed in TBST and incubated over night with the appropriate primary antibodies (1/1000 dilution) in TBST containing 3% BSA and 0.1% sodium azide (Sigma-Aldrich) at 4 °C and in a shaker. After three washes with TBST, the membranes were incubated 1 h with horseradish peroxidase-conjugated anti-rabbit or anti-mouse IgG at 1/2500 dilutions in TBST containing 5% nonfat dry milk. Immunoreactive material was visualized by ECL.

### 2.10. Multiplex Cytokine ELISA Array

MDA-MB-231 cells were serum-starved for 24 and 48 h. Then, conditioned media was collected, clarified by centrifugation, and stored at −20 °C until further evaluations. The relative abundance of the secreted cytokines was determined using a Multiplex Human Cytokine ELISA Kit (MyBioSource, San Diego, CA, USA) and following the manufacturer’s instructions. The optical density (OD) values of the samples were obtained at 450 nm.

### 2.11. Immunoprecipitation Procedures

Protein A beads (30 μL slurry) were co-incubated overnight under rotation at 4 °C in 4 mL of TNBC conditioned media (CM), with either the anti-IL-6 mAb diluted 1/100 or the IgG isotype control (2 µg). Next, each mixture was centrifuged, the supernatants collected, filtered through a 0.2 µm filter, and frozen until further evaluation. The beads were boiled with Laemmli buffer and applied to a 7.5% SDS-PAGE alongside the supernatants, then transferred to PVDF membranes and immunoblotted to determine the efficiency of the immunoprecipitation.

### 2.12. Chemotactic Cell Migration Assay

Cell migration assays were carried out using the Real-Time Cell Analyzer (RTCA) Dual-Plate (DP) Instrument of the xCELLigence system (Roche Diagnostics, Basel, Switzerland). Adherent ADMSC monolayers were trypsinized and seeded (30,000 cells/well) onto CIM-Plates 16 (Roche Diagnostics). These migration plates are similar to conventional transwells (8 µm pore size) but have gold electrode arrays on their bottom side of the membrane to provide real-time data acquisition of cell migration. Prior to cell seeding, the underside of the wells from the upper chamber were coated with 25 µL of 0.15% gelatin in PBS and incubated for 1 h at 37 °C. Cell migration was continuously monitored for 8 h using serum-free, CM obtained from MDA-MB-231 cells (CM or TNBC cells secretome) grown in the presence or absence of EGCG. Serum-free media was used as a cell migration negative control (NM). Basal migration experiments consisted of pre-treating the cells with NM or CM +/− EGCG 10 μM for 24 h and then allowing cells to migrate without chemoattractant (NM). In all cases, the impedance values were measured by the RTCA DP Instrument software and were expressed as Normalized Cell Migration Index. Each experiment was performed two times in triplicates.

### 2.13. Statistical Data Analysis

Data and error bars were expressed as mean ± standard error of the mean (SEM) of three or more independent experiments unless otherwise stated. Hypothesis testing was conducted using the Kruskal–Wallis test followed by a Dunn Tukey’s post-test (data with more than 3 groups) or a Mann–Whitney test (two group comparisons). Probability values of less than 0.05 (*) or 0.01 (**) were considered significant and denoted in the figures. All statistical analyses were performed using the GraphPad Prism 7 software (San Diego, CA, USA).

## 3. Results

### 3.1. Transcriptomic Analysis of Human Adipocyte-Derived Mesenchymal Stem/Stromal Cells Response to Variations of the TNBC Secretome

In order to first identify the genes and molecular pathways involved in the acquisition of a CAA phenotype, human ADMSC were cultured for 24 h in conditioned media (CM) isolated from serum-starved TNBC-derived MDA-MB-231 cells (TNBC cells secretome). Total RNA was isolated as described in the Methods section and submitted to RNA-Seq. We found a total of 13,284 genes differentially expressed in both conditions (FDR < 0.05), from which roughly two thirds were downregulated in the presence of EGCG (Figure 1A). Next, we selected genes with at least a two-fold change expression variation with respect to the control and a maximum corrected *p*-value of 0.001 to perform gene and pathway enrichment analysis. Using the previous criteria, we found 1331 differentially expressed genes (DEGs), among which 107 were up-regulated and 1224 were down-regulated. The DEGs were cross-referenced with actual protein–protein interactions to build active-subnetworks, onto which pathway enrichment analysis was performed to decipher which biological pathways were enriched in response to the TNBC cells secretome. Among the pathways involved in the adaptive response of ADMSC to the TNBC cells secretome, nine reached statistical significance below a *p*-value < 0.01 (Figure 1B, left panel). The following ones were highlighted: nuclear factor-kappa B (NF-κB) signaling, cytokine-cytokine receptor interaction, pathways related to cytokine intracellular signaling such as the tumor necrosis factor (TNF)-α and transforming growth factor (TGF)-β, insulin resistance, breast cancer, central carbon metabolism in cancer, HIF-1α, and the interaction between advance glycation end products (AGE)- and their receptors (RAGE). Interestingly, the highlighted pathways have a significant degree of connectivity, with upregulated IL-6, IL-1, and other soluble cytokines acting as network hubs (Figure 1C). In addition, ADMSC exposed to the TNBC cells secretome acquired a pro-inflammatory phenotype. To validate the CAA- and immunomodulatory-related genes found in our RNA-Seq experiment, an RT^2^ Profiler RT-qPCR gene array was performed. The TNBC cells secretome induced more than 10-fold the expression of CAA genes identified from the RNA-Seq analysis. These included CCL2, CCL5, IL-1β, IL-6, and vascular endothelial growth factor alpha (VEGFα) (Figure 1D). Other upregulated immunomodulatory genes included cytokines with a chemotactic role, such as C-X-C Motif Chemokine Ligand 5 and 8 (CXCL5 and CXCL8), and C-C motif ligand 20 (CCL20). In addition, mediators of the inflammatory response also included cyclooxygenase 2 (COX2), indoleamine 2,3-dioxygenase (IDO), programmed death-ligand 1 (PD-L1), IL-1β, and IL-6. These results confirm the DEG found in our transcriptomic analysis and demonstrate the effective induction of a CAA-like phenotype in ADMSC in response to the TNBC cells secretome at the gene expression level.

### 3.2. EGCG Inhibits the Expression of Biomarkers Associated with the CAA Phenotype, Epithelial-to-Mesenchymal Transition, and Inflammatory Signaling Pathways Induced by the TNBC Cell Secretome

Once the increase in expression of genes linked to a CAA phenotype was demonstrated, we analyzed the effect of EGCG on this adaptive response. ADMSC were incubated 24 h with the TNBC cells secretome in the presence of 10 μM EGCG, vehicle (ethanol), or negative media (NM). After harvesting the cells and extracting the total RNA, we performed RNA-Seq and analyzed the output, as described in the Methods section. The cells incubated with NM were used as a control group. Gene and pathway enrichment analysis of the samples challenged with EGCG showed few common pathways compared to the control (Figure 1B, right). Then, we restricted our analysis to those genes with a corrected *p*-value < 0.001 and an absolute FC greater than two. A Venn diagram depicting the percentage and absolute number of DEGs found in each condition was shown (Figure 1E, left). The cells receiving EGCG had six times more DEGs than the vehicle-treated cells compared to the NM-treated cells. From the 8818 genes modulated by EGCG, 8070 DEGs were unique to these samples, while 748 were present in both conditions. Our next goal was to identify and characterize the expression pattern of common DEGs. We performed a robust k-means clustering, identifying seven clusters based on the gene expression distribution in the samples (Figure 1E, right). Genes from clusters 1–4 are downregulated in both samples, and their intra-cluster differences result from the magnitude and pattern of downregulation. On the other hand, cluster 6 contained genes switched-on in both EGCG and vehicle-treated samples compared to NM-treated samples (Appendix A). The most attractive genes were clustered in groups 5 and 7, with antagonizing expression patterns associated with CM. The EGCG inhibited the genes assigned to cluster 5 (Table 1), which, according to gene ontology (GO) enrichment analysis using an FDR < 0.05 corresponded to growth factors, intracellular signaling molecules and modification enzymes with acyltransferase activity (Appendix A). The genes listed in Table 1 evidenced the antagonist effect of EGCG over the CM-mediated induction of genes associated with cholesterol and lipids biogenesis (HMGCS1, HMGCR, IDI1, STARD4, GPAM, and ACSL4), proliferation (PID1, BMP6, and FGF7), invasion and metastasis (PLOD2, MMP1, CEMIP2, PAPPA, COL8A1, and ADAM12), glucose transport (STEAP1, STEAP2), cell survival and oncogenesis (CCN4, WNT5A, and FGF7), and vesicular trafficking (RAB27B, TRFC). Surprisingly, the genes from cluster 7 could not be associated statistically with a particular protein class or molecular functions. More detailed information on all shared genes is provided as a supplementary EXCEL data sheet (Appendix A).

Following the transcriptomic RNA-Seq analysis, RT-qPCR was performed to validate the inhibition of key CAA markers in ADMSC upon treatment with the TNBC cells secretome in the presence or absence of 10 μM EGCG. The addition of EGCG reduced or completely abrogated the induction of CCL2, CCL5, CXCL8, IL-1β, IL-6, VEGFα, HIF-1α, COX2, and IDO (Figure 2A, black bars), while it did not affect that of PD-L1. In addition to altering ADMSC gene expression plasticity, we assessed the chemotactic response of ADMSC to TNBC cells secretome by a real-time cell migration assay. An increased ADMSC migration index was observed in response to the direct exposure to TNBC cells secretome (Figure 2B, black circles), or when ADMSC were cultured for 24 h with it (conditioned ADMSC) and then left to migrate without chemoattractant (Figure 2C, black circles). EGCG prevented the induced chemotactic (Figure 2B) or acquired response (Figure 2C) in both scenarios.

### 3.3. The Epithelial-to-Mesenchymal Transition (EMT) Contributes to the CAA-Induced Phenotype in ADMSC

We found a robust induction of the transcription factors Slug and Snail and of the pro-inflammatory cytokine IL-6 at the protein level (Figure 3A). As expected, EGCG reduced their expression by at least 50% (Figure 3B). Complementing our transcriptomic results, we also found a CM-dependent activation of signaling cascades involving the phosphorylation of NF-κB and Smad2 transcription factors (Figure 3A), which was reduced in the presence of EGCG (Figure 3B).

### 3.4. Snail as a Crucial Intermediate in the Upregulation of CAA Genes in Response to TNBC Cells Secretome

Since incubation of the ADMSC with the TNBC cells secretome triggered the EMT biomarker Snail expression, we investigated its contribution to regulating other CAA genes. We silenced Snail in ADMSC using siRNA (siSnail) and then exposed these cells to the TNBC cells secretome for 24 h. Our siRNA efficiently reduced Snail transcript and protein expression (Figure 4A,B) and impaired the production of CAA genes, except for VEGFα and HIF-1α (Figure 4C). This suggests that selective Snail-mediated transcriptional control is involved in the transcriptional regulation of CAA genes in ADMSC in response to TNBC cells secretome.

### 3.5. The TNBC-Derived MDA-MB-231 Secrete High Levels of IL-6

Our bio-informatics analysis revealed a strong activation of the cytokine-receptor interaction pathway in the ADMSC response to the TNBC cells secretome (Figure 1). This suggests a contribution of the cytokines secreted by the TNBC cell-derived MDA-MB-231 to the phenotypic modification and chemotaxis of ADMSC. Using a protein cytokine array, we aimed to identify the main cytokines present in the CM upon 24 and 48 h of serum starvation. IL-6 was identified as the preponderant cytokine present in the CM, followed by VEGFα, and to a lesser extent IL-1β, epithermal growth factor (EGF), and transforming growth factor beta (TGFβ) (Figure 5A). Interestingly, exogenous addition of IL-6 to ADMSC triggered a biphasic chemotactic dose-response, with the maximal migration rate observed at 10 pg/mL (Figure 5B). This suggests that ADMSC are responsive to both the autocrine and paracrine effects of IL-6.

### 3.6. The IL-6 Secreted by the TNBC-Derived MDA-MB-231 Is Required for the Chemotactic Response of the ADMSC but Does Not Trigger the CAA Phenotype

To test the role of IL-6 in the chemotactic response of the ADMSC to the TNBC cells secretome, we depleted the CM from IL-6 by immunoprecipitation (CM, Figure 6A). This inhibited ADMSC chemotaxis as we compared the chemotactic capacity of IL-6 immunoprecipitated from the TNBC cells secretome (Figure 6B, black triangles) to that of the CM immunoprecipitated with an IgG isotype control (Figure 6B, black circles). In another approach, IL-6 was silenced in TNBC-derived MDA-MB-231 cells (Figure 6C, left), then CM harvested upon 24 h of incubation with serum-deprived culture media. Reduced IL-6 concentration was confirmed by ELISA (Figure 6C, right), and the reduced level of IL-6 negatively impacted the ADMSC chemotactic response (Figure 6D). Altogether, these results suggest that IL-6 exerts a significant role in ADMSC mobilization in response to TME cues.

Lastly, we inquired the extent to which IL-6 contributed to the paracrine upregulation of Snail, autocrine regulation of IL-6, and the signal-transducing pathways activated upon the response to TNBC cells secretome. ADMSC were thus treated for 24 h with IL-6 at 10 pg/mL, a concentration corresponding to its maximal chemotactic effect, and at 10 ng/mL that corresponded to the naturally occurring concentration of IL-6 in the TNBC cells secretome (data not shown). Cells were harvested and protein lysates prepared to detect the CAA signature. Unexpectedly, neither Snail nor IL-6 were induced by both of the IL-6 concentrations tested (Figure 7A). However, when ADMSC were incubated with IP-CM (IgG isotype control vs. anti-IL-6 mAb), the IL-6 depleted-CM still induced Snail and IL-6 expression as well as activated the NF-κB and Smad2 pathways to the same extent as the control IP-CM did (Figure 7B). This suggests that, although Snail controls the acquisition of a CAA phenotype and that IL-6 is involved in the chemotactic response of ADMSC (Figure 6B), unknown factors within the TNBC cells secretome, beyond IL-6, are involved in the acquisition of the CAA phenotype in ADMSC.

## 4. Discussion

The TME is a highly dynamic niche composed of multiple cell types constantly interacting with each other. Accordingly, tumor cells can recruit not only immune cells but also those cells residing from adjacent tissues like the adipose tissue [36]. The adipose tissue is predominant within the breast anatomy, and its endocrine functions recognized to support tumor cells at their early stages of malignant transformation [37,38]. To understand the role of adipocytes in tumor progression, previous studies have co-cultured human breast cancer cells with human mature adipocytes [6,24,39], leading to an increase in the invasiveness of the cancer cells, and the onset of a pro-inflammatory state characterized by the induction of cytokines such as IL-6, IL-1β, and CCL5 [24,39]. Histological analysis of human breast tumors has shown that the adipocytes present at their invasive front also expressed high levels of MMP-11 and IL-6, in contrast to adipocytes from healthy tissues where these proteins are absent [6]. This suggests a transducing mechanism that transitions from a healthy adipocyte to a CAA phenotype. Therefore, our findings support that TNBC cells can also mobilize and promote the reprogramming of undifferentiated ADMSC with inflammatory and CAA-like phenotypes. This finding is particularly relevant in the context of obesity, where the abnormal extent of the adipose tissue provides a substantial source of inflammation cytokines for neoplasms [40], and where ADMSC as well as early-stages of adipocytes maturation may also contribute.

Here, the TNBC cells secretome triggered significant chemotaxis in ADMSC and induced a pro-inflammatory phenotype characterized by the overexpression of IL-1β, COX2, VEGFα, and IL-6 among other immunomodulators, as well as a CAA phenotype including the induction of CCL2 and CCL5. These results were further supported herein by the identification of signalling pathways such as NF-κB, HIF-1α, and AGE-RAGE, all induced as highlighted from the bio-informatics analysis, and where the upregulation of IL-6 connected pathways was modulated by obesity-like insulin resistance, TNF signaling, AGE-RAGE, and cytokine-receptor interaction (Figure 1). Such evidence at early stages of adipocyte maturation is supported by data showing that ADMSC were primed and permanently altered by tumor presence in breast tissue, resulting in increased tumor cell invasiveness [41]. Interestingly, our data support these studies as the EMT biomarker Snail is induced in ADMSC in response to the soluble factors present in the MDA-MB-231 CM. More importantly, Snail induction was proved to be essential for sustained upregulation of IL-6, IL-1β, CCL2, and CCL5.

Despite its abundance in the TNBC cells secretome and its clear contribution to the ADMSC chemotaxis, exogenous addition of IL-6 failed to switch on its autocrine upregulation of Snail. Remarkably, EGCG acted as a potent inhibitor of the pro-inflammatory state triggered by the TNBC cells secretome, generating a robust inhibition of CAA markers and pro-inflammatory cytokines. This role for EGCG was partially attributed to reduced induction of Snail. Furthermore, EGCG decreased the chemotactic potential of the TNBC cells secretome and inhibited the activation of the NF-κB and Smad2 signalling cascades.

The pathway involving HIF-1α, classically activated by low oxygen concentrations (hypoxia), upregulates the expression of pro-angiogenic and mitogenic cytokines such as leptin and VEGFα while reducing the levels of the antimitogenic adipokine adiponectin [42]. Harvesting of the TNBC-cells-derived secretome through cell culture serum starvation is typically reflected by enriched lactate levels, which has been proposed to mimic a biochemical “perception” of hypoxia regardless of the level of oxygen, leading to the secretion of angiogenic and inflammatory growth factors/cytokines [43]. Besides the increased HIF-1α observed in ADMSC, our transcriptomic screen identified the AGE-RAGE pathway, which may play an important role in acquiring the CAA phenotype; oxidative stress; and activation of the Smad2 and NF-κB signaling, thus leading to the secretion of inflammatory cytokines and growth factors [44,45,46]. Interestingly, adipose tissue inflammation occurs in obesogenic conditions due to hypoxia and is thought to originate from enlarged adipocytes distant from the vasculature [47]. Furthermore, hypoxia has now been directly demonstrated to occur in adipose tissue of several obese mouse models (*ob/ob*, KKAy, diet-induced) and to lead to increased HIF-1α levels [42,48].

Our experimental approach exploits the paracrine up-regulation of CCL2, CCL5, IL-1β, and IL-6 in ADMSC in response to TNBC cells secretome, confirming and complementing prior co-culture approaches mixing cancer cells and adipocytes [6,24]. In addition, relevant cytokines for acquiring a tumor malignancy phenotype, such as CCL5/RANTES, have also been shown to increase cell motility and invasiveness in high-glucose culture conditions [24]. Interestingly, such conditions mimic the adaptive cellular responses triggered during the onset of insulin resistance associated with obesity. Although we still need validation at protein levels for all the genes found in our transcriptomic analysis, those including IL-6, IL-1β, CCL2, CXCL2, and HIF-1α were present in our dataset and bridged several pathways associated with pro-tumoral roles.

On the other hand, we validated the autocrine induction of IL-6 in ADMSC, both at gene and protein levels. Paradoxically, despite ADMSC producing IL-6 after incubation with the TNBC cells secretome, this was not a direct response to paracrine IL-6. One must conclude that irrespective of its massive levels in the CM, the IL-6 paracrine induction certainly requires a combination of stimuli. Noteworthily, the production of IL-6 by ADMSC exposed to the TNBC cells secretome depended on Snail expression. This signaling interplay between Snail and IL-6 has been proposed in myofibroblast trans-differentiation during oral submucosal fibrosis, a premalignant disorder of the oral cavity [49].

Once we identified the main upregulated genes and pathways involved in the acquisition of the CAA phenotype in ADMSC in response to the TNBC cells secretome, we investigated the impact of EGCG. This catechin is known to modulate molecular targets and signaling pathways associated with cell survival, proliferation, differentiation, migration, angiogenesis, hormone activities, detoxification enzymes, and immune response [50,51,52,53,54]. Our differential transcriptomic analysis performed in ADMSC treated in the presence or absence of EGCG revealed six times more modulated genes in samples treated with EGCG (92%, 8070 DEGs). The presence of EGCG drastically reduced the expression of genes encoding growth factors, intracellular signaling intermediates, and cytokines, all of which prevented the acquisition of a CAA-like phenotype.

## 5. Conclusions

The present study revealed EGCG’s ability to inhibit the chemotactic properties of the TNBC cells secretome, primarily through NF-κB and Smad2 signal-transducing pathways, suggesting that a diet-derived intervention could efficiently alter the signaling crosstalk that links TNBC cells to the CAA phenotype within the adipose tissue environment. Most importantly, our study presents evidence that EGCG can efficiently target the CAA-like phenotype of ADMSC and prevent the onset of a TME that would favor breast cancer development.

## Figures and Tables

**Figure 1 nutrients-14-01099-f001:**
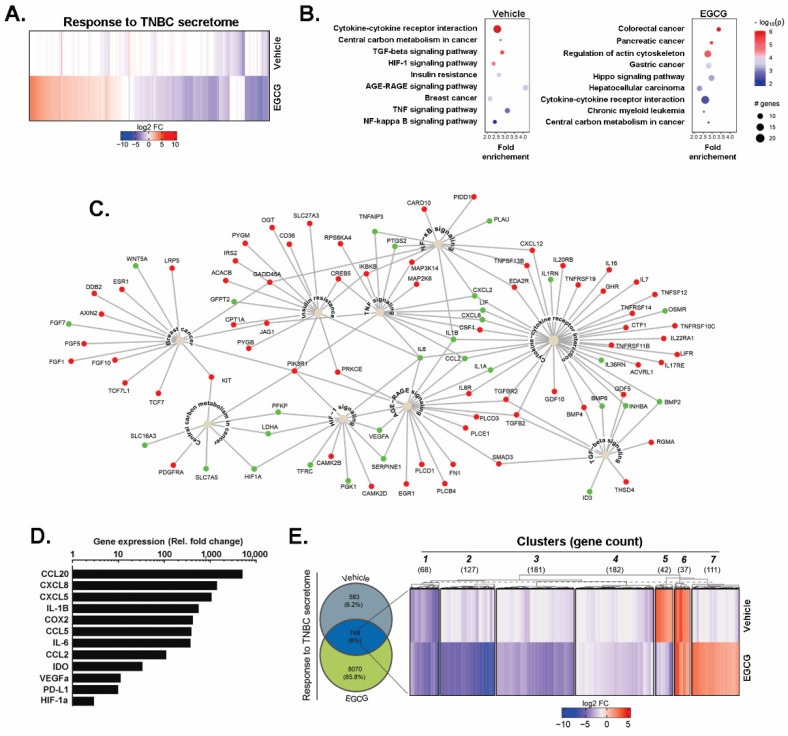
Transcriptomic modulation of ADMSC(adipose-derived mesenchymal stem/stromal cells) response to variations of the TNBC cell secretome. Human pre-adipocytes mesenchymal stem/stromal cells (ADMSC) were incubated for 24 h with serum-free media conditioned for 48 h by TNBC-derived MDA-MB-231 cells (TNBC cells secretome) in the presence of 10 μM EGCG(epigallocatechin gallate) or Ethanol (vehicle). Total RNA was extracted from triplicate samples, and gene expression modulation was assessed through RNA-Seq analysis, as described in the Methods section. (**A**) Fold change gene expression compared to control cells considered below the significant threshold (FDR < 0.05) in the presence or absence of EGCG. (**B**) KEGG pathways enrichment analysis of differentially expressed genes (DEGs) with an absolute fold change (FC) > 2 and adjusted *p*-value < 0.001 in each experimental condition. (**C**) Network graph showing enriched pathways and their respective DEGs in the absence of EGCG. Upregulated and downregulated genes are color-coded in green and red, respectively. (**D**) Expression of selected genes associated with cancer-associated adipocyte (CAA) phenotype and immunomodulation was confirmed by RT-qPCR, as described in the Methods section using a Human Cancer Inflammation and Immunity Crosstalk RT^2^-Profiler gene array kit. (**E**) Venn diagram showing the number of DEGs in each experimental condition, followed by a robust k-means clustering visualized as a heatmap. Number of genes per cluster is shown in parenthesis. Representative genes shared in both conditions with a log_2_FC > 2 and *p*-value < 0.001. CCL20 (C-C motif ligand 20); CXCL8 and CXCL5 (C-X-C Motif Chemokine Ligand 8 and 5); IL-1β and IL-6 (interleukin 1 beta and 6); COX2 (cyclooxygenase 2); CCL5 and CCL2 (chemokine C-C motif ligand 5 and 2); VEGFa (vascular endothelial growth factor alpha); IDO (indoleamine 2,3-dioxygenase); HIF1a (hypoxia inducible factor 1 alpha) and PD-L1 (programmed death-ligand 1).

**Figure 2 nutrients-14-01099-f002:**
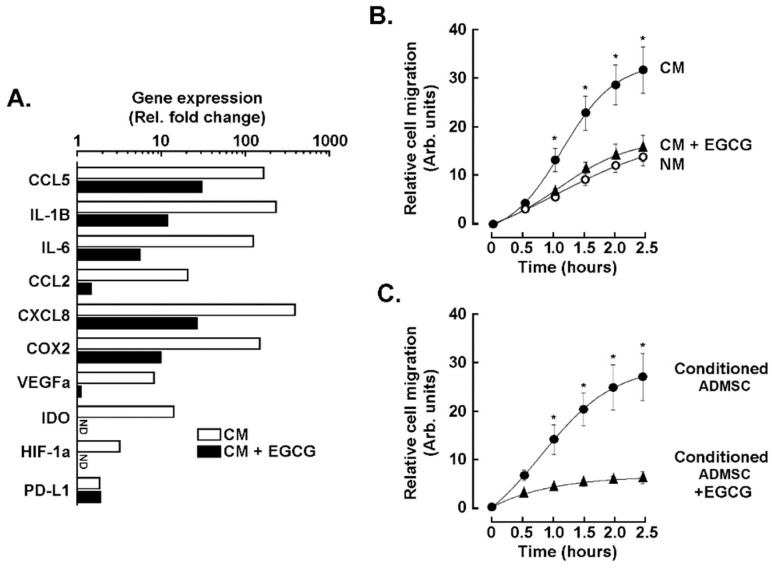
EGCG alters the acquisition of a CAA(cancer-associated adipocyte) phenotype and chemotactic response. (**A**) ADMSC response to TNBC cells secretome was monitored after 24 h in vehicle-treated cells (white bars), or in the presence of 10 μM EGCG (black bars). Total RNA was isolated, cDNA was synthetized, and CAA genes induction was evaluated using a RT^2^-Profiler RT-qPCR gene array kit. A representative experiment out of two screens is shown. (**B**) Relative cell migration rate of ADMSC in response to TNBC cells secretome (CM, closed circles), CM with 30 μM of EGCG (closed triangles), or serum-free negative media (NM, open circles). (**C**) Basal cell migration response: ADMSC were treated for 24 h with CM (closed circles) or in the presence of CM supplemented 10 μM EGCG (closed triangles); then, basal cell migration was assessed. Data are representative from two independent experiments performed in triplicate. (ND, not detectable). Statistical differences were determined with a Mann–Whitney two tail test with a *p* < 0.05 (*). CCL5 and CCL2 (chemokine C-C motif ligand 5 and 2); IL-1β and IL-6 (interleukin 1 beta and 6); CXCL8 (C-X-C motif chemokine ligand 8); COX2 (cyclooxygenase 2); VEGFa (vascular endothelial growth factor alpha); IDO (indoleamine 2,3-dioxygenase); HIF1a (hypoxia inducible factor 1 alpha) and PD-L1 (programmed death-ligand 1).

**Figure 3 nutrients-14-01099-f003:**
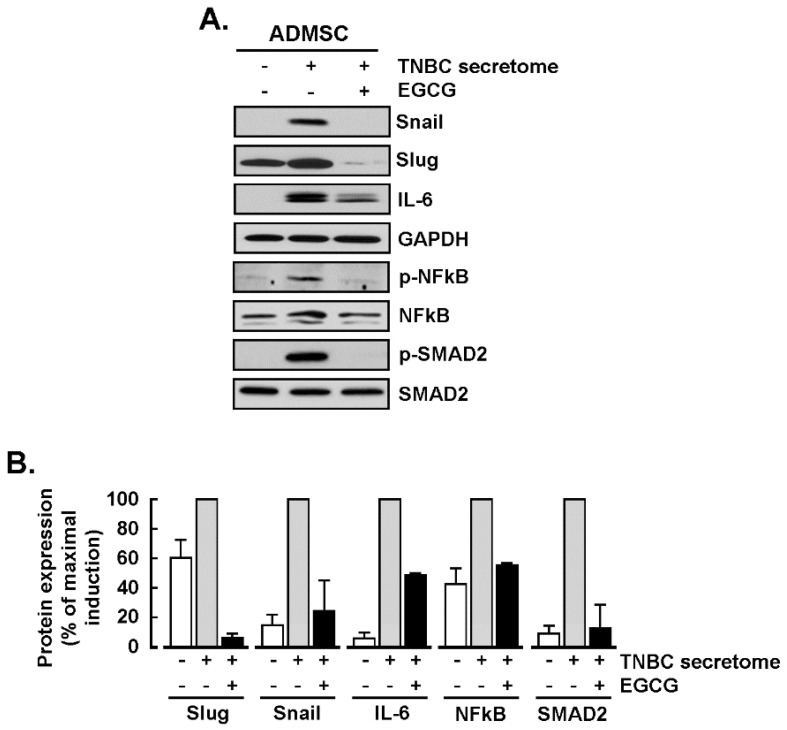
EGCG inhibits the induction of the pro-inflammatory cytokine IL-6, epithelial-to-mesenchymal transition (EMT) markers, and NF-κB and Smad2 signal transducing pathways. ADMSC were incubated for 24 h with the TNBC cells secretome and protein lysates collected, as described in the Methods section for Western blotting. (**A**) Immunoblotting of Snail, Slug, IL-6, and the phosphorylated and total forms of NF-κB and Smad2 (20 μg of protein/well). (**B**) Representative densitometric analysis of Snail, Slug, IL-6, and the ratio of phosphorylated/total forms of NF-κB and Smad2. Data are expressed as the percent of maximal effect for each marker in ADMSC treated with the TNBC cell secretome (grey bars). Cells treated with negative media (NM, white bars) and TNBC cells secretome in the presence of 10 μM EGCG (black bars). Data are representative of three independent experiments. Snail (Snail family transcriptional repressor 1); Slug (Snail family transcriptional repressor 2); IL-6 (Interleukin 6); NFκβ (nuclear factor κβ); SMAD2 (mothers against decapentaplegic homolog 2); GAPDH (glyceraldehyde 3-phosphate dehydrogenase).

**Figure 4 nutrients-14-01099-f004:**
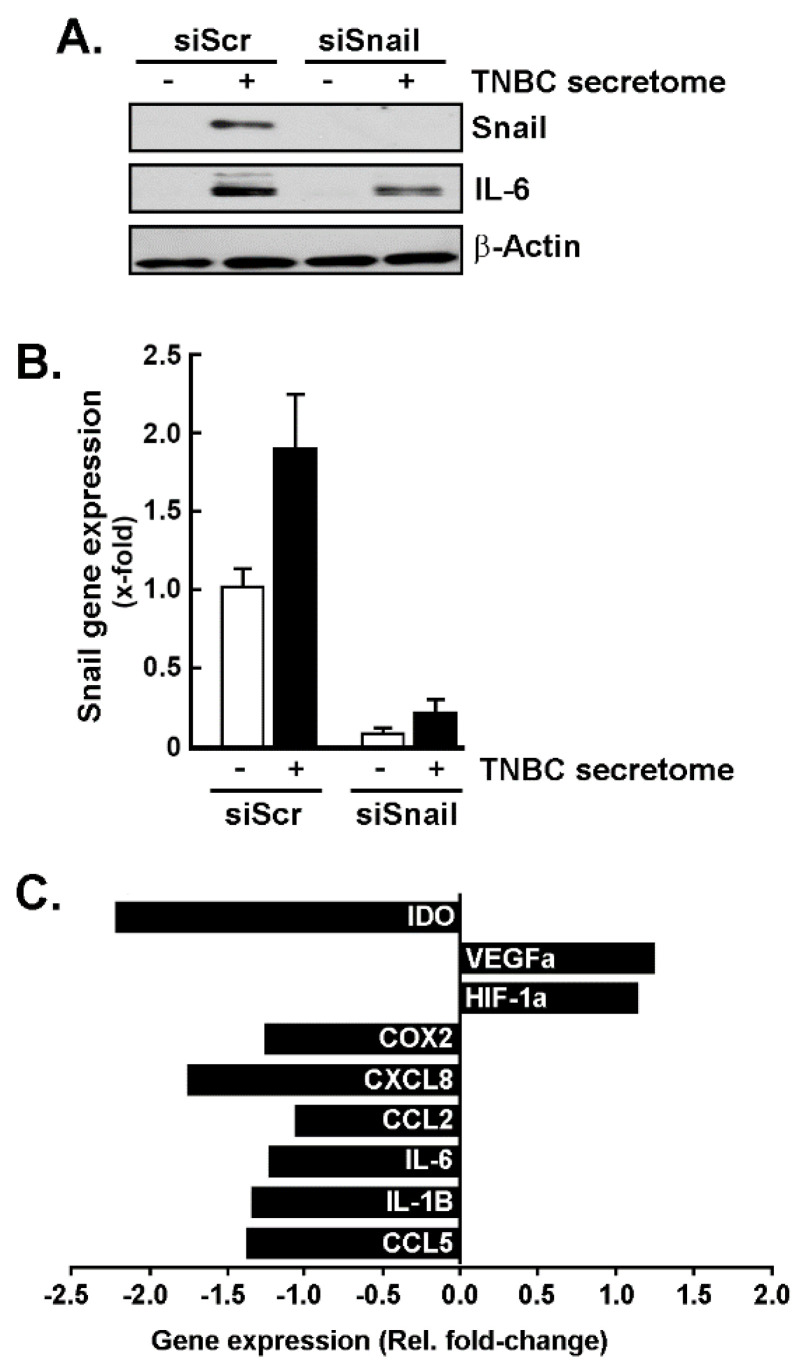
Role of Snail in the upregulation of the CAA phenotype genes. Transient gene silencing of Snail (siSnail) or of control (siScrambled) was performed in ADMSC, followed by an incubation with the TNBC cells secretome for 24 h. Cell lysates and total RNA were isolated, and levels of protein and gene expression assessed by Western blotting and RT-qPCR, respectively. (**A**) Protein levels of Snail and IL-6 were assessed by immunoblotting in ADMSC transfected with siScr or siSnail. (**B**) Snail gene expression was evaluated by RT-qPCR in ADMSC transfected with siScr or siSnail, and treated with TNBC cell secretome (black bars) or negative media (white bars). (**C**) CAA gene expression resulting from the comparison of ADMSC transfected with siSnail in response to the TNBC cells secretome vs. siScr cells incubated with the same CM (reference group).

**Figure 5 nutrients-14-01099-f005:**
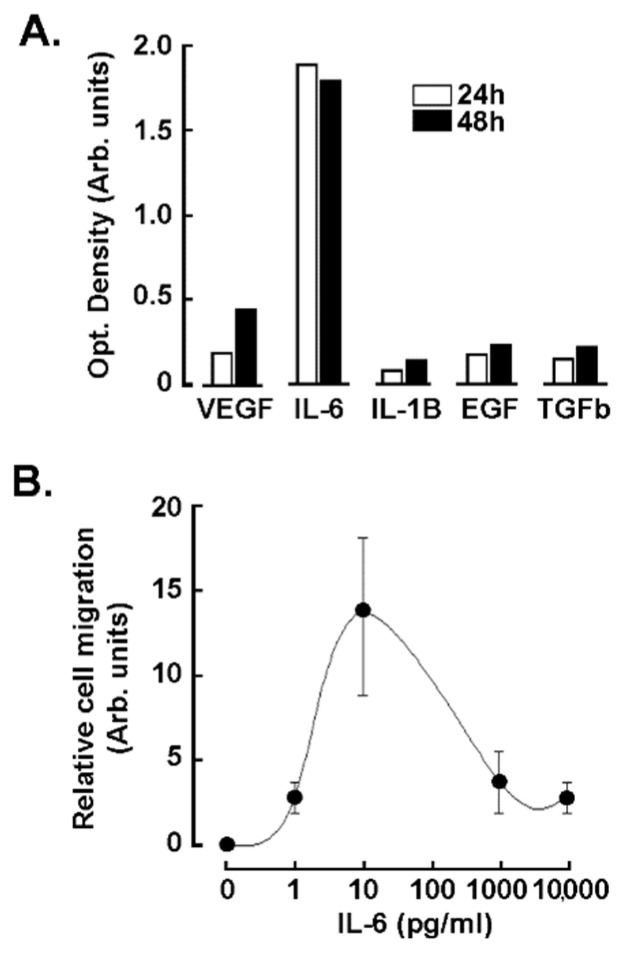
Cytokines levels present in the TNBC cells secretome. (**A**) Human TNBC-derived MDA-MB-231 cells were cultured for 24 h (white bars) and 48 h (black bars) in serum free media, and then their respective secretome collected and the cytokines concentrations determined using an ELISA array as described in the Methods section. (**B**) IL-6-mediated chemotaxis of ADMSC was assessed in real time, as described in the Methods section using the exCELLigence system (one out of three independent experiments is shown).

**Figure 6 nutrients-14-01099-f006:**
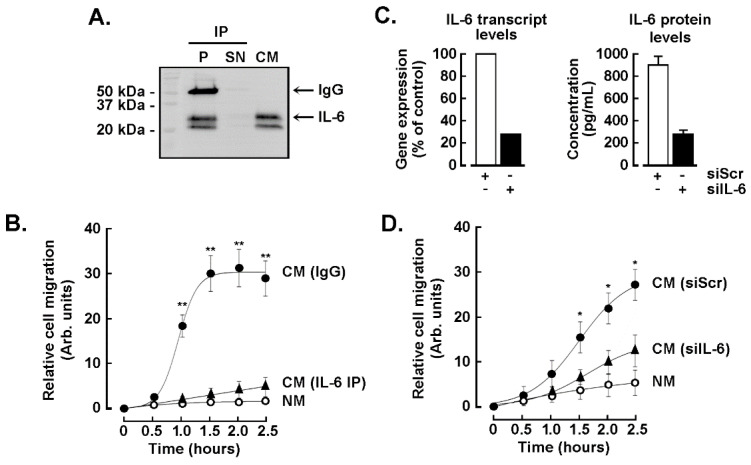
Role of IL-6 in the ADMSC chemotactic response to the TNBC cells secretome. MDA-MB-231 cells were cultured in serum free media for 48 h and the conditioned media harvested (TNBC cells secretome). (**A**) Immunoprecipitation (IP) of IL-6 from the TNBC cells secretome was performed as described in the Methods section. The efficiency of the IP was evaluated by immunoblotting of the pellet (P), supernatant (SN), or of the conditioned media before the IP (CM). IgG indicates the heavy chain of the anti-IL-6 antibody used for the IP. (**B**) Chemotactic response of the ADMSC in response to negative media (open circles), to CM upon control IgG isotype IP (closed circles), or to CM upon anti-IL-6 IP (closed triangles). (**C**) Transient gene silencing of IL-6 was performed in MDA-MB-231 cells as described in the Methods section. Control cells were transfected with siRNA-Scrambled (siScr). Cells were then serum starved for 24 h. Total RNA was extracted, and RT-qPCR performed to monitor IL-6 silencing efficiency (left). CM was collected to assess secreted IL-6 levels using an ELISA. (**D**) ADMSC chemotactic response to TNBC cells secretome was monitored in CM harvested from siScr-transfected MDA-MB-231 cells (closed circles), CM harvested from siIL-6-transfected cells (closed triangles), or in response to negative media (NM, open circles). One out of two independent experiments performed in triplicate is shown. Statistical differences were determined with a Mann–Whitney two tail test with a *p* < 0.05 (*) or *p* < 0.01(**).

**Figure 7 nutrients-14-01099-f007:**
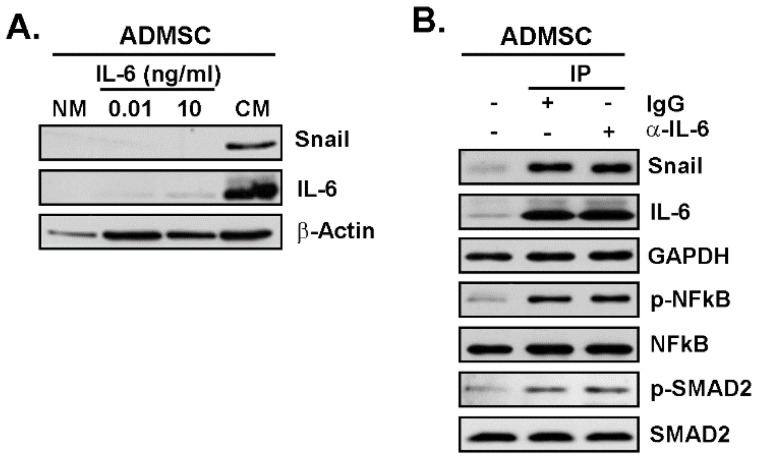
IL-6 is not sufficient to trigger the CAA phenotype in ADMSC. (**A**) ADMSC were incubated for 24 h with negative media (NM), TNBC cell-derived secretome (MDA-MB-231 conditioned media, CM), or 10 pg/mL or 10 ng/mL IL-6 in NM. Cell lysates were next isolated and protein expression of Snail, IL-6, and β-Actin assessed by immunoblotting. (**B**) Immunoblots showing the phosphorylation status of Smad2 and NF-κB, and the expression of Snail and IL-6 in cells treated with NM, IL-6-IP depleted media (αIL-6), or IgG isotype-IP control media (IgG). Data are representative of one experiment out of two.

**Table 1 nutrients-14-01099-t001:** The effect of EGCG(epigallocatechin gallate) over Cluster 5 genes upregulated by the TNBC secretome.

ENSEMBL	Gene	Fold Change Vehicle vs. Control	Fold Change EGCG vs. Control	Gene Description	Enriched Terms by KEGG Analysis
ENSG00000122641	INHBA	5.36	−2.51	Follicle-Stimulating Hormone-Releasing Protein/secreted	- TGF-beta signaling pathway - Signaling pathways regulating pluripotency of stem cells - Cytokine-cytokine receptor interaction - Associated with cancer cachexia in human patients
ENSG00000152952	PLOD2	4.51	−4.00	2 procollagen-lysine/cisternae of the RER.	- Collagen formation and degradation of the extracellular matrix - Oxidoreductase activity
ENSG00000170961	HAS2	4.25	−2.99	Hyaluronan synthase 2 Polysaccharide/extracellular matrix	- Glycosaminoglycan metabolism - Hyaluronan synthase activity
ENSG00000105835	NAMPT	3.94	−1.54	Nicotinamide phosphoribosyltransferase; enzime	- NOD-like receptor signaling pathway - Cytokine with immunomodulating properties - Adipokine with anti-diabetic properties - Stress response
ENSG00000104321	TRPA1	3.87	−4.76	Transient receptor potential cation channel, subfamily a/transmembrane proteins	- Regulation of TRP channels - Signal transduction - Growth control
ENSG00000112972	HMGCS1	3.73	−2.37	3-α-hydroxy-3-methylglutaryl-coa synthase 1/	- PPAR signaling pathway - Terpenoid backbone biosynthesis - Cholesterol and lipid homeostasis
ENSG00000196611	MMP1	3.68	−1.80	Matrix metalloproteinase 1/interstitial collagenase	- PPAR signaling pathway - Relaxin signaling pathway - Calcium ion binding - Metallopeptidase activity
ENSG00000067064	IDI1	3.56	−2.33	Isopentenyl-diphosphate delta isomerase 1/peroxisomally-localized enzyme	- Terpenoid backbone biosynthesis - Regulation of cholesterol biosynthesis - mTOR signalling
ENSG00000041353	RAB27B	3.40	−6.85	Member RAS oncogene family/membrane-bound proteins involved in vesicular fusion and trafficking	- Vesicular fusion and trafficking - Autophagy pathway and metabolism of proteins - GTP binding and protein domain specific binding
ENSG00000104415	CCN4	3.39	−2.87	Wnt1-inducible signaling pathway protein 1/	- WNT1 signaling pathway - Associated with cell survival
ENSG00000157214	STEAP2	3.39	−2.92	Six-transmembrane epithelial antigen of prostate/metalloreductase localized in Golgi complex, plasma membrane, and in the cytosol.	- Mineral absorption - Transport of glucose and other sugars, bile salts, and organic acids
ENSG00000164211	STARD4	3.14	−2.38	Start domain-containing protein 4	- Metabolism of steroid hormones - Lipid binding
ENSG00000119927	GPAM	3.13	−4.02	Glycerol-3-phosphate acyltransferase/Mitochondrial	- Regulation of cholesterol biosynthesis and triacylglycerol biosynthesis. - Acyltransferase activity
ENSG00000120437	ACAT2	3.11	−2.60	Acetyl-coa acetyltransferase 2/cytosolic	- Terpenoid backbone biosynthesis - Acyltransferase activity
ENSG00000171208	NETO2	3.09	−1.57	Neuropilin- and tolloid-like 2/transmembrane protein	- Ionotropic glutamate receptor binding in the brain
ENSG00000153823	PID1	3.00	−1.64	Phosphotyrosine interaction domain-containing 1	- Proliferation of preadipocytes without affecting adipocytic differentiation
ENSG00000153162	BMP6	2.89	−2.05	Bone morphogenetic protein 6	- Secreted ligand of the TGF-beta superfamily of proteins - Growth factor activity
ENSG00000164647	STEAP1	2.80	−2.36	Six-transmembrane epithelial antigen of prostate 1/cell surface antigen significantly expressed at cell-cell junctions.	- Mineral absorption (copper homeostasis) - Glucose/energy metabolism - Oxidoreductase activity and channel activity
ENSG00000135048	CEMIP2	2.76	−2.24	Cell migration inducing hyaluronidase 2/transmembrane protein	- Regulator of angiogenesis and VEGF signaling
ENSG00000113161	HMGCR	2.70	−2.13	3-@hydroxy-3-methylglutaryl-coa reductase/rate-limiting enzyme for cholesterol synthesis	- AMPK signaling pathway - Terpenoid backbone biosynthesis. - Cholesterol and lipid homeostasis - Protein homodimerization activity and NADP binding
ENSG00000114251	WNT5A	2.70	−2.10	Wingless-type mmtv integration site family/secreted signaling proteins	- Wnt signaling pathway - Hippo signaling pathway - Hepatocellular carcinoma - Signaling pathways regulating pluripotency of stem cells - Breast Cancer - Basal cell carcinoma - Cushing syndrome - DNA-binding transcription factor activity and protein domain specific binding - Oncogenesis
ENSG00000182752	PAPPA	2.67	−1.72	Pregnancy-associated plasma protein/secreted metalloproteinase	- Metabolism of proteins - Regulation of Insulin-like Growth Factor (IGF) transport - Metalloendopeptidase activity
ENSG00000125845	BMP2	2.61	−2.26	Bone morphogenetic protein 2/regulatory element: cis-acting enhancer	- Protein heterodimerization activity - Cytokine activity
ENSG00000144810	COL8A1	2.60	−2.67	Collagen, type VIII	- Migration and proliferation of vascular smooth muscle cells
ENSG00000140416	TPM1	2.40	−3.17	Tropomyosin 1/actin-binding proteins involved in the contractile system of muscles	- Hypertrophic cardiomyopathy - Dilated cardiomyopathy - Cytoskeletal protein binding
ENSG00000068366	ACSL4	2.34	−5.44	acyl-coa synthetase long chain family	- PPAR signaling pathway. - Lipid biosynthesis and fatty acid degradation
ENSG00000140285	FGF7	2.30	−5.16	Fibroblast growth factor 7	- Regulation of actin cytoskeleton - Gastric Cancer - Breast cancer - Rap1 signaling pathway - Ras signaling pathway - MAPK signaling pathway - Melanoma - Growth factor activity - Chemoattractant activity
ENSG00000072274	TFRC	2.07	−2.52	Transferrin receptor/ cell surface receptor	- HIF-1 signaling pathway - Receptor-mediated endocytosis - Clathrin derived vesicle budding
ENSG00000100644	HIF1A	2.05	−5.07	Hypoxia-inducible factor 1, alpha subunit	- HIF-1 signaling pathway - Central carbon metabolism in cancer - PD-L1 expression and PD-1 checkpoint pathway in cancer - Kaposi sarcoma-associated herpesvirus infectio - Th17 cell differentiation - Thyroid hormone signaling pathway - Proteoglycans in cancer - Choline metabolism in cancer - Autophagy - Renal cell carcinome
ENSG00000148848	ADAM12	2.02	−3.54	A disintegrin and metalloproteinase domain 12	- Cell-cell and cell-matrix interactions - Metallopeptidase activity

KEGG: Kyoto Encyclopedia of Genes and Genomes; RER: rough endoplasmic reticulum; NOD: nucleotide-binding oligomerization domain; PPAR: peroxisome proliferator-activated receptors; mTOR: mechanistic target of rapamycin; GTP: guanine nucleotide-binding proteins; WNT1: wingless-type MMTV integration site family, member 1; TGF-beta: transforming growth factor beta; Rap1: Ras-proximate-1; MAPK: mitogen-activated protein kinase.

## Data Availability

All data generated or analyzed during this study are included in this published article.

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
