# Peer review of "EGCG Prevents the Onset of an Inflammatory and Cancer-Associated Adipocyte-like Phenotype in Adipose-Derived Mesenchymal Stem/Stromal Cells in Response to the Triple-Negative Breast Cancer Secretome"

_nutrients, 2022, doi:10.3390/nu14051099_

Round 1

Reviewer 1 Report

In their study, the authors report on the effects of ECGC on the cancer-associated phenotype in adipose-derived stromal cells in response to the secretome of triple-negative breast cancer cells. Deciphering the effect of breast cancer cells on cells in their direct environment (mainly consisting of adipose tissue resident cells including adipose-derived stem cells) and agents that affect this interaction is important to understand and modulate the mutual interaction between breast cancer and adipose tissue. The present study makes a valuable contribution in this context.

However, there is a general and some specific points that should be clarified.

General:

The authors investigated in detail the effects of breast-cancer cell conditioned medium on the phenotype of adipose-derived stromal cells (ASCs). Based on the results of gene and protein expression studies, a phenotype of cancer-associated adipocytes was attributed to the ASCs. In their conclusion, the authors describe this as a “CAA phenotype in the early stages of adipocytes maturation”. The association with adipocytes appears throughout the manuscript.

In the study, however, the ASCs did not undergo adipogenic differentiation and no evidence for a “adipocyte” phenotype is provided, so in the reviewer’s opinion, the term “adipocyte” or “cancer-associated adipocyte” is not correct and misleading here as well as the attribution of a “early stage of differentiation” or the statement that “ECGC can efficiently target early-stage adipocyte maturation”.   

There are studies investigating the impact of breast cancer cells on the phenotype of ASCs. Here, the aquisition of a myofibroblast or cancer-associated fibroblast phenotype is associated with ASC within the breast cancer microenvironment (see Chandler et al. 2012, PNAS 209, p. 9786-9791, Song et al 2017, Matrix Biology 60-61, p. 190-205 and others). A study of Senst et al. (2013, Breast Cancer Res Treat 137, 69-79) used, in part, a similar approach to investigate the impact of the breast cancer cell secretome on the migration of ASCs.

In my opinion, the introduction and discussion should refer more specifically to ASC in the tumor context and include relevant studies.

Specific:

What is the rationale behind the chosen ECGC concentrations (effect on ASC gene and protein expression 10 µm, effect in ASC migration 30 µM)?

In figure 6B,  transient snail-silencing in ASCs is shown by qRT-PCR. What are the „arbitrary units“ representing snail gene expression?

Figure 6C shows the downregulation of „CAA gene expression“ in siSnail-transfected ASCs in response to the TNBC secretome. Are these data from a PCR array or qRT-PCR analysis and what is the „Rel. fold-change“ related to? Please specify!

The ELISA cytokine array is not described in Material and Methods. Please depict the array membranes, at least in the supplement.

Fig 1B is not completely shown, Table 1 not completely depicted.

Supplemental 1 consists of a exel sheet listing regulated genes, whereas the legend speaks of A) „pie charts“ and B) fold enrichment values for protein class and molecular functions.

Supplemental 2 is missing

Author Response

Question #1 : This reviewer states that “The present study makes a valuable contribution in the context of [mutual interaction between breast cancer and adipose tissue];

Answer : We thank this reviewer for the appreciation.

Question #2 : This reviewer suggests that we better define the cell model used in this study.

Answer : We do agree with this reviewer as no “adipocyte” phenotype analysis was performed here. We have now better defined the used adipose-derived mesenchymal stem/stromal cells in the Introduction and Methods sections as follows :

“ Nevertheless, the cues triggering the CAA phenotype remain less understood at the early stages of adipocyte maturation as well as in adipocyte-derived mesenchymal stem/stromal cells (ADMSC, also referred to as pre-adipocytes). Those cells have been demonstrated to have the ability to differentiate into mesodermal tissue lineages including adipose through the regulation of key transcriptional factors involved in early adipogenesis.”

“ The human adipose-derived mesenchymal stem/stromal cells (ADMSC) and TNBC-derived cell line MDA-MB-231 were purchased from the American Type Culture Collection (ATCC, Manassas, VA). ADMSC were grown in Mesenchymal Stem Cell Basal Medium (ATCC, PCS-500-030), and supplemented with Mesenchymal Stem Cell Growth Kit - Low Serum (ATCC, PCS-500-040). They were defined as having the property, in the 1st days of adipogenic induction, to upregulate key transcriptional factors during early adipogenesis.”

Question #3 : This reviewer suggests that we address the cancer-associated fibroblast phenotype of our ADMSC with regards to ASC.

Answer : We agree with this reviewer that some studies have shown that the breast cancer microenvironment triggered the acquisition of a myofibroblast or cancer-associated fibroblast phenotype as reflected through IL-8-mediated Fibronectin induction. In our current ADMSC model, no such induction of Fibronectin was detected in response to TNBC secretome as compared to those inductions observed for other CAA genes and proteins. IL-8 was also not significantly detected in the TNBC secretome as assessed using ELISA.

Question #4 : This reviewer wishes that we provide the rationale behind the EGCG concentrations used.

Answer : We found that 10 mM EGCG was sufficient to explore the long term effects on protein expression at 24 hours. On the other hand, 30 mM EGCG was found optimal to monitor short term effects such as those on cell chemotaxis assessed at 2.5 hours.

Question #4 : This reviewer wishes that we define “arb. units” in Fig.5B.

Answer : We thank this reviewer for the careful review. We have now corrected the y-axis labeling as “x-fold”.

Question #5 : This reviewer wishes that we specify whether the data from Fig.4B are from a PCR array or from qRT-PCR analysis, and to define the “rel x-fold change” better.

Answer : We apologize for the confusion. Fig.4B data originate from qRT-PCR array analysis, and relative x-fold change refers to the extent of EGCG effect in the responsiveness to TNBC secretome.

Question #6 : This reviewer noticed that the ELISA cytokine array was not described in the Methods section.

Answer : Such description was now added as follows :

“ 2.10. Multiplex cytokine ELISA array : MDA-MB-231 cells were serum-starved for 24 and 48 hours. Then, conditioned media was collected, clarified by centrifugation and stored at -20°C until further evaluations. The relative abundance of the secreted cytokines was determined using a Multiplex Human Cytokine ELISA Kit (MyBioSource, San Diego, CA), and following the manufacturer’s instructions. The optical density (OD) values of the samples were obtained at 450 nm. ”

Question #7 : This reviewer mentions that Fig.1B and Table.1 were not completely depicted, and that Supplemental-2 data was missing.

Answer : We have now re-drawn and extensively described Fig.1. As for Table.1, we believe that it contains sufficient descriptors to allow the reader to asses the nature of the genes modulated.

Question #8 : This reviewer noticed that the EXCEL sheet of Supplemental-1 is not correctly labeled in the corresponding legend.

Answer : This reviewer confuses the legend of the Supplemental.1 figure (that was originally correct) vs the original EXCEL data sheet which has now been labeled as Supplemental.3

Reviewer 2 Report

In the research paper entitled “EGCG prevents the onset of a cancer-associated adipocyte phenotype in adipose-derived mesenchymal stem cells in response to the triple-negative breast cancer secretome”, the authors tried to show the effect of EGCG on altering the cancer-associated adipocyte phenotype. The experiments are well done but the authors should highlight a few points:

  1. Figure1 is so abruptly presented. The authors should perform GSEA and DAVID analysis and show the enriched pathways. Then they should show that most of the enriched pathways are related to inflammatory and other cancer-related gene signatures. From that, they should highlight the genes and their foldchange in the data using a heat map.
  2. Then, if CAA has any genes related to its phenotype (other than the common inflammatory genes like IL6 and certain chemokines) and which got highlighted in their RNA seq data, authors should perform real-time PCR to validate the RNA seq data along with highlighting the CAA phenotype.
  3. In Figure2: they should show the usage of EGCG and highlight the altered transcriptomic changes using GSEA. At the end of figure two, the authors should present the before EGCG and after EGCG altered genes (important for CAA phenotype in the Before-After graph).
  4. The authors should rewrite the paper as the paper has a lot of grammatical mistakes.
  5. The authors should also improve the presentation of results. One or two graphs in a panel are too few. They should combine a few figures and make a big panel.

Author Response

Question #1 : This reviewer states that “The experiments are well done [in this research paper]”.

Answer : We thank this reviewer for the appreciation.

Question #2, #3, and #4 : This reviewer suggests that, in Fig.1, we 1) rather show the enriched pathways, and analyze there relation to inflammation and cancer-related gene signatures, 2) that the differentially highlighted genes then be validated by qPCR, and that 3) we highlight the altered transcriptomic changes in Fig.2 then present the before vs after EGCG effects.

Answer : We thank this reviewer for the suggestion. A heat map has now been performed from which 120 pathways were found enriched, and 90 of them were selected. We further have included more data from our bioinformatics analysis addressing this point. Fig.1 now shows all the genes deemed statistically significant after DESeq2 (Panel A) in both conditions. This provides a better perspective to the reader about the nature of the changes induced by EGCG.

Moreover, our analysis with the pathfindeR package uses similar approaches to classical GSEA and DAVID, with the bonus of leveraging this analysis with curated protein-protein interaction data. We have included the corresponding pathway and gene enrichments for both conditions (Panel B). The gene signature is included in Table.1. We also have made clear in the text that the qPCR data presented in Figure 1 (Panel D) is the validation step for the gene signature. Precisely, the common chemokines and interleukins are central to the studied phenomenon.

Finally, we envisioned this paper as a validation stage of the often controversial data available in that topic. Therefore, we planned to focus primarily on soluble factors and transcription factors. We have a set of genes previously unknown to this, and which play a role in this cellular crosstalk phenomena. These are out of the scope of this study, but will be included in a follow-up paper.

Question #5 : This reviewer suggests that we pay attention for grammatical mistakes, and suggests that we envision combining some figures.

Answer : We have now gone through the manuscript and corrected/edited as much as possible the text to the best of our knowledge. Any further comments will be welcomed. We have also combined the original Fig.1 and Fig.2, to generate a new Fig.1. The same was made to Fig.4 and Fig.5, to generate a new Fig.3.

Round 2

Reviewer 2 Report

The authors responded to all the queries raised and modified the manuscript accordingly.